# Survey of Australian Dietitians Contemporary Practice and Dietetic Interventions in Overweight and Obesity: An Update of Current Practice

Erin D. Clarke [1,2], Rebecca L. Haslam [1,2], Jennifer N. Baldwin [3], Tracy Burrows [1,2], Lee M. Ashton [1,4] and Clare E. Collins [1,2,*]

1   School of Health Sciences, College of Health, Medicine and Wellbeing, The University of Newcastle, Newcastle, NSW 2308, Australia
2   Food and Nutrition Research Program, Hunter Medical Research Institute, New Lambton Heights, NSW 2305, Australia
3   Institute for Musculoskeletal Health, The University of Sydney and Sydney Local Health District, Sydney, NSW 2000, Australia
4   School of Education, College of Human and Social Futures, The University of Newcastle, Newcastle, NSW 2308, Australia
*   Correspondence: clare.collins@newcastle.edu.au; Tel.: +61-2-4921-5646

**Abstract:** The aim of this survey was to identify, (1) use of Dietitians Australia best practice guidelines, (2) contemporary practices/knowledge, and (3) professional development needs of Australian dietitians in relation to management of clients with overweight or obesity. A cross-sectional online survey consisting of 67 multiple choice and Likert scale questions on the three survey aims was administered. Data were collected between 2020–2021 and reported descriptively as frequency (%). Of 178 survey attempts, 80 respondents completed all questions (45%). Most respondents spent >50% of their time working with individuals with overweight/obesity, usually in private practice (47%). Two thirds of respondents had accessed best practice guidelines, but only 12% had completely read best practice guidelines. General healthy eating was the most frequent dietary approach used (77%). Seventy-five percent (n = 56) of dietitians reported that dietary interventions were selected based on client preference. Almost half of dietitians rated their knowledge and level of skill in management of obesity as good. Approximately 60% (n = 41) dietitians reported their skill gap was related to providing behavioural therapy/counselling. Results of the current survey indicate that use of best practice guidelines is low. However, dietitians surveyed reported that they had a good understanding of obesity management and choose a client centred approach to management, which is in line with current recommendations. Professional development activities, particularly regarding behavioural counselling are of particular interest to dietitians working with individuals with obesity.

**Keywords:** dietitian; best practice; weight management



## 1. Introduction

Obesity is a risk factor for chronic disease, including type 2 diabetes and cardiovascular disease [1]. In 2018, the Australian Institute of Health and Welfare reported that 38% of the national burden of disease could be prevented through change in modifiable risk factors, including nutrition and body weight which accounted for 5.4% and 8.4%, respectively [2]. Accredited Practising Dietitians (APDs) play an important role in treatment for individuals with overweight and obesity [3–5]. Evidence shows that weight loss is greater following dietitian-led interventions compared to those receiving usual care from another health professional or health program [3]. Three prior Australian surveys assessed dietetic service provision, intervention strategies and professional development needs regarding management of adults and children with overweight and obesity in 1997 [6], 2002 [4] and 2011 [5]. Key findings from these surveys included the need for best practice dietetic

guidelines for management of overweight and obesity [4] and greater awareness of the obesity management guidelines [5]. Considering it has been ten years since the last survey and with changes in research findings and potential impact on practice, there is a need to update survey findings.

Since the implementation of the 2011 survey [5], the National Health and Medical Research Council (NHMRC) developed the 2013 Clinical Practice Guidelines for the Management of Overweight and Obesity [7]. These have since been rescinded further highlighting the need for revised guidelines. Similarly, the 2012 Dietitians Australia Best Practice Guidelines for the Treatment of Overweight and Obesity in Adults are outdated [8]. While recent management guidelines are not available in Australia, in 2020 Obesity Canada released the evidence-based Canadian Adult Obesity Clinical Practice Guidelines [9], informed by recent evidence reviews including assessment of obesity, medical treatment and approaches for medical nutrition therapy and physical activity.

Dietitians Australia, the dietetics accreditation body in Australia, made a public call to action to update Australian clinical practice guidelines in early 2021 through a media release [10]. In working towards an update of best practice, an evaluation into current dietetic management of overweight and obesity is timely. Therefore, the aim of the current survey was to identify use of best practice guidelines, contemporary practices and knowledge, and professional development needs of Australian dietitians in relation to management of clients with overweight or obesity.

## 2. Materials and Methods

The online survey was conducted as an update following three previous surveys in 1997 [6], 2002 [4] and 2011 [5]. The current survey used the same questions as the previous surveys and contained 67 questions, divided into three domains; (1) demographic and service profiles of respondents, (2) the current use of the 2012 Dietitians Australia Best Practice Guidelines for the Management of Overweight and Obesity, including professional development and (3) current dietetic practice, including a question on future research directions, detailed below. The survey targeted dietetic professionals who could work in any area (for example clinical, private practice but not limited to these) involved in providing medical nutrition therapy for people with overweight or obesity. Non-dietitians, students and retired clinicians were not eligible. The survey was distributed via a survey link administered through Qualtrics XM System (Provo, UT, USA) by Dietitians Australia and Dietitian Connection through member weekly email updates. These avenues were chosen as Dietitians Australian and Dietitian Connection are the major membership groups of Australian dietitians, with the aim to reach as many practising dietitians as possible. Due to COVID-19 disruptions the survey was distributed on two separate occasions, initially in 2020 and again in 2021. No incentives were offered for completing the survey. Ethics approval for this survey was granted by the University of Newcastle Human Research Ethics Committee (H-2020-0212). These results have been reported using STROBE guidelines (Supplementary Table S1).

Demographic and service profiles of respondents were reported through 12-items, for example years as an APD, proportion of time working with people with overweight and obesity. For full survey see Supplementary Table S2.

Previous utilisation of the Dietitians Australia Best Practice Guidelines for the Management of Overweight and Obesity was assessed through 9-items that asked about access, use, change in practice arising from the guidelines or attendance of any continued professional development activities on the best practice guidelines (Supplementary File S2 Questions 13–21).

The third survey domain consisted of 46-items to assess current practice. This included questions on caseload, referrals, intervention strategies, frequency of interventions, evaluation of practice and patient outcomes, enablers and barriers to evidence-based practice, future continued professional development and research recommendations. Questions were closed with an "other, please specify" option available to capture additional responses or used a Likert scale to rank respondent's answers. For example, 5-point Likert scales were

used to rank knowledge from "poor" (1-point) to "excellent" (5-points) with lower points suggesting poorer knowledge. Some questions used frequency of respondents' level of agreement with questions rated from "strongly agree" to "strongly disagree" (Supplementary File S2). Some questions used throughout the survey allowed multiple responses, these questions are marked with a superscript letter corresponding with a footnote explaining this in the tables presented in the results.

Best practice weight management was assessed from 33-items within the third survey domain that asked how often respondents assessed certain outcomes or used specific strategies with clients when managing overweight or obesity. These responses were ranked on a 5-point Likert scale from "never" to "always". Total scores were generated based on methods outlined previously [4,5], with questions scoring a possible 0–2 points for each response and a maximum score of 62-points in total.

*Statistical Analysis*

Results were presented by the categories of 'completers only', 'non-completers' and for all respondents. Completers were defined as respondents who completed the survey and non-completers were defined as those who started the survey but did not make it to the end of the survey. Questions were not compulsory therefore numbers do not always add up to 100%. Written results are reported for completers only unless otherwise stated. The majority of data presented were categorical and therefore reported as frequency (%). Chi-squared tests were undertaken to evaluate any differences in demographic characteristics between the categories of completers versus non-completers, with Fisher's exact test used to determine non-random associations. A sensitivity analysis by year of survey completion was conducted using chi-squared. One way ANOVA with Bonferroni post hoc comparisons were used to test differences in best practice weight management scores across sub-groups (years of practice, area of practice, and time spent working with individuals with obesity, year of survey completion). Further associations between best practice scores and enablers, barriers and gaps in knowledge related to obesity management were examined using ANOVA. Cronbach's alpha was used to assess the internal consistency and reliability of the questions used to generate the best practice scores. Statistical analyses were conducted using Stata version 14.2 (StataCorp, College Station, TX, USA), $p < 0.05$ was considered statistically significant.

## 3. Results

### 3.1. Demographics and Service Profiles of Respondents

The survey was attempted by 178 participants; 43 in 2020 and 135 in 2021. In total 80 respondents completed the full survey (45%), Figure 1. Demographic profiles of completers compared to non-completers were not significantly different, with the exception of membership status where non-completers were all dietitians but not all APDs ($p = 0.047$). Of the respondents who completed the survey, the majority resided in Victoria (n = 22, 29%) or New South Wales (n = 21, 28%), (Table 1).

Most respondents (n = 47, 63%) worked in metropolitan or large urban areas, while less than 10% (n = 6) worked in rural areas. Approximately half the respondents had been a practising dietitian for <5 years and two thirds were working full-time (>20 h/week). Most respondents managed patients with overweight and obesity in private practice (n = 35, 47%) and worked in a tertiary treatment/prevention service (n = 55, 75%). Approximately a third of respondents (64%) were not involved in any obesity interest groups. Of respondents who were involved in an interest group the most commonly reported was the Dietitians Australia obesity interest group. Half (n = 40) reported that their service used clinical practice guidelines for obesity management (Table 1).

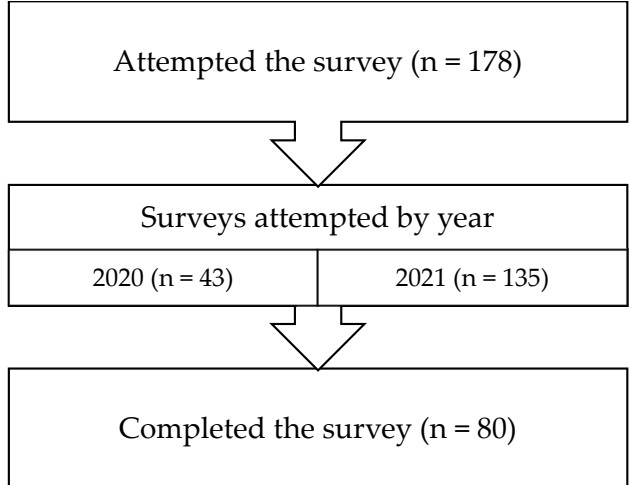

**Figure 1.** Breakdown of survey completion.

**Table 1.** Survey Respondents' Demographics.

| Question | Responses | Completers (n = 80) N (%) | Non-Completers (n = 98) N (%) | Chi-Squared *p*-Value |
|---|---|---|---|---|
| **Which Dietitians Australia Branch do you belong to?** | Number of responses | 75/80 | 80/98 | 0.26 |
| | Northern Territory | 0 (0) | 2 (2.0) | |
| | Queensland | 16 (21.3) | 12 (12.2) | |
| | New South Wales | 21 (28.0) | 24 (24.5) | |
| | Victoria | 22 (29.3) | 24 (24.5) | |
| | South Australia | 5 (6.7) | 2 (2.0) | |
| | Tasmania | 1 (1.3) | 0 (0) | |
| | Western Australia | 8 (10.7) | 6 (6.1) | |
| | Australian Capital Territory | 0 (0) | 4 (4.1) | |
| | Overseas | 2 (2.7) | 4 (4.1) | |
| **What is your membership status?** | Number of responses | 75/80 | 78/98 | 0.047 |
| | Member with dietetic qualifications and APD | 75 (100) | 74 (75.5) | |
| | Member with dietetic qualifications and no APD | 0 (0) | 4 (4.1) | |
| **How would you describe your geographical location?** | Number of responses | 75/80 | 78/98 | 0.16 |
| | Metropolitan or large urban area | 47 (62.7) | 59 (60.2) | |
| | Regional | 22 (29.3) | 13 (13.3) | |
| | Rural/remote | 6 (8.0) | 6 (6.1) | |
| **How many years have you been a practising dietitian?** | Number of responses | 75/80 | 78/98 | 0.09 |
| | <5 years | 37 (49.3) | 23 (23.5) | |
| | 5–10 years | 11 (14.7) | 24 (24.5) | |
| | 11–15 years | 7 (9.3) | 12 (12.2) | |
| | >15 years | 20 (26.7) | 19 (19.4) | |
| **Are you currently working?** | Number of responses | 74/80 | 78/98 | 0.08 |
| | Full time (>20 h/week) | 47 (63.5) | 53 (54.0) | |
| | Part time (up to 20 h/week) | 27 (36.5) | 21 (21.4) | |
| | Not currently working | 0 (0) | 4 (4.1) | |

**Table 1.** *Cont.*

| Question | Responses | Completers (n = 80) N (%) | Non-Completers (n = 98) N (%) | Chi-Squared *p*-Value |
|---|---|---|---|---|
| **What is your current employment status?** | Number of responses | 75/80 | 78/98 | |
| | Employed in Australia as a dietitian | 70 (93.3) | 62 (63.3) | |
| | Employed in Australia but not in nutrition or dietetics | 1 (1.3) | 2 (2.0) | 0.22 |
| | Overseas | 2 (2.7) | 6 (6.1) | |
| | Not employed but looking for work | 0 (0) | 2 (2.1) | |
| | Not employed and not looking for work | 0 (0) | 1 (1.0) | |
| | Other | 2 (2.7) | 5 (5.1) | |
| **In what area of practice do you manage clients with overweight and obesity?** | Number of responses | 75/80 | 78/98 | |
| | Community nutrition | 15 (20.0) | 18 (18.4) | |
| | Government department/NGO | 3 (4.0) | 3 (3.1) | |
| | Public Hospital | 15 (20.0) | 8 (8.2) | |
| | Private hospital | 1 (1.33) | 3 (3.1) | 0.43 |
| | Aged care | 0 (0) | 2 (2.1) | |
| | Private practice | 35 (46.7) | 34 (34.7) | |
| | Research/education | 0 (0) | 1 (1.0) | |
| | Other | 6 (8.0) | 9 (9.2) | |
| **What proportion of work do you spend time working in the area of overweight or obesity?** | Number of responses | 75/80 | 78/98 | |
| | <10% | 7 (9.3) | 10 (10.2) | |
| | 10–25% | 12 (16.0) | 13 (13.3) | |
| | 26–50% | 20 (26.7) | 22 (22.4) | 0.21 |
| | 51–75% | 26 (34.7) | 15 (15.3) | |
| | 76–100% | 10 (13.3) | 18 (18.4) | |
| **Do you work with other members of a multidisciplinary team?** | Number of responses | 74/80 | 16/98 | |
| | Yes | 63 (85.1) | 15 (15.3) | 0.36 |
| | No | 11 (14.9) | 1 (1.0) | |
| **What other services are provided within your multidisciplinary team?** [(a),(b)] | Number of responses | 62/63 | 10/15 | |
| | Psychologist | 34 (54.8) | 7 (7.1) | 0.37 |
| | Physiotherapist | 37 (59.7) | 4 (4.1) | 0.24 |
| | Gym instructors | 3 (4.8) | 1 (1.0) | 0.51 |
| | General Practitioner | 31 (50.0) | 6 (6.1) | 0.56 |
| | Exercise physiologists | 33 (53.2) | 6 (6.1) | 0.69 |
| | Social workers | 18 (29.0) | 0 (0) | 0.049 |
| | Other | 24 (38.7) | 3 (3.1) | 0.60 |
| **Are you a member of any obesity interest groups (Dietitians Australia or other)?** | Number of responses | 74/80 | 78/98 | |
| | No | 47 (63.5) | 38 (38.8) | 0.07 |
| | Yes | 27 (36.5) | 40 (40.8) | |
| **What obesity interest groups are you apart of?** [(a),(b),(c)] | DA National Obesity Interest Group | 18 (72.0) | 24 (24.5) | 0.95 |
| | ANZOS | 2 (8.0) | 2 (2.0) | 0.77 |
| | OSSANZ | 3 (12.0) | 3 (3.1) | 0.72 |
| | Other | 5 (20.0) | 7 (7.1) | 0.91 |
| **Does your service use the clinical guidelines for obesity management?** | Number of responses | 75/80 | 75/98 | |
| | Yes | 40 (53.3) | 36 (36.7) | |
| | No | 23 (30.7) | 18 (18.4) | 0.19 |
| | Unsure | 12 (16.0) | 21 (21.4) | |

**Table 1.** *Cont.*

| Question | Responses | Completers (n = 80) N (%) | Non-Completers (n = 98) N (%) | Chi-Squared *p*-Value |
|---|---|---|---|---|
| **What year did your service start using the clinical guidelines for obesity management?** [c] | Number of responses<br>1980–2000<br>2001–2011<br>2012–2017<br>2018–2020 | 22/40<br>0 (0)<br>2 (9.2)<br>12 (54.6)<br>8 (36.4) | 12/36<br>1 (1.0)<br>1 (1.0)<br>4 (4.1)<br>6 (6.1) | 0.40 |

ANZOS—Australia and New Zealand Obesity Society; APD—Accredited Practising Dietitian; NGO—Non-Government Organisation; OSSANZ—Obesity Surgery Society of Australia and New Zealand. [a]—multiple responses allowed and therefore results may be added to greater than 100%. [b]—responses scored as a 'yes' or 'no'. [c]—only answered by respondents who answered "yes" to the prior question.

### 3.2. Use of Dietitians Australia Best Practice Guidelines for the Management of Overweight and Obesity

Approximately 12% of completers (n = 9) reported that they had accessed and completely read the guidelines, here 22% (n = 40) of all respondents did not answer this question and stopped the survey. Twenty-eight percent (n = 20) of respondents reported they made changes to their practice after reading the guidelines, with a change in how they managed and treated clients the most commonly reported change (Supplementary Table S3).

### 3.3. Current Dietetic Practice

Caseloads were predominately adult females and males (56% and 33%, respectively). Only 21 respondents reported that their workplace provided a specialised obesity service, of those 21 respondents 70% (n = 16) reported the service was within a specialist medical service including bariatric surgery (n = 9), specialist obesity clinic (n = 8), type 2 diabetes (n = 5) and endocrine clinic (n = 5).

The most common philosophical approach of respondents' obesity services was a combination of diet, exercise and behaviour (n = 61, 82%). The most frequent dietary approaches used by services was general healthy eating (n = 57, 77%). Dietary strategies or interventions were selected for clients mostly based on client preference (n = 56, 76%), Table 2. The majority of respondents worked within a multidisciplinary team (n = 63, 85%), most frequently alongside psychologists (n = 34, 55%) and physiotherapists (n = 37, 60%), Table 1.

Most frequently respondents reported that the number of times a client was reviewed before being discharged after an initial consult was "variable/dependent on client needs" (n = 34, 46%), with time intervals for follow up between 2 weeks and 1 month (n = 41, 55%). Half of respondents selected that clients' progress was "often" monitored for >6 months (n = 35, 49%). Almost all respondents monitored client progress through diet improvements (n = 70, 95%), Table 2. Approximately a third of respondents' services (n = 23) had a protocol, policy or clinical pathway for the dietetic management of patients with overweight or obesity. At the time of the survey only 34% (n = 25) of respondents reported evaluating the effectiveness of different dietary interventions within their service, most commonly through dietary improvements (Table 2) with few reporting quality audits such as structure and outcomes. A sensitivity analysis was conducted to compare the approached used by year of completion, the only significant difference found was in the use of meal replacements as a strategy for patients (Supplementary Table S4).

**Table 2.** Dietary strategies and monitoring of outcomes used by respondents.

| | | Completers Only (n = 80) | All Respondents (n = 91) |
|---|---|---|---|
| | | N (%) | N (%) |
| **Services dietary approach [a]** | Number of responses | 74/80 | 91/91 |
| | General healthy eating | 57 (77.0) | 68 (74.7) |
| | Set energy level plan | 21 (28.4) | 25 (27.5) |
| | Formulated meal plan | 12 (16.2) | 17 (18.7) |
| | General advice on low fat eating | 14 (18.9) | 18 (19.8) |
| | Specific low fat eating plan | 0 (0) | 2 (2.2) |
| | Non-diet approach with specific focus to reduce energy intake | 40 (54.1) | 49 (53.9) |
| | Non-diet approach eating behaviour goals | 42 (56.8) | 55 (60.4) |
| | Very low energy diet | 32 (43.2) | 38 (41.8) |
| | Meal replacements | 18 (24.3) | 20 (22.0) |
| | Health at Every Size | 26 (35.1) | 36 (39.6) |
| | Other | 6 (8.1) | 7 (7.7) |
| **How specific dietary strategies or interventions selected for clients [a]** | Number of responses | 74/80 | 91/91 |
| | Client preference | 56 (75.7) | 68 (74.7) |
| | Client past dieting experience | 41 (55.4) | 51 (56.0) |
| | Dietitian practitioner experience | 35 (47.3) | 39 (42.9) |
| | Based on program/service philosophy | 9 (12.2) | 11 (12.1) |
| | As requested by medical referral | 14 (18.9) | 17 (18.7) |
| | Method negotiated by practitioner with client | 50 (67.6) | 61 (67.0) |
| | Other | 4 (5.4) | 4 (4.4) |
| **Client outcome measures used to monitor progress (up to 6 months follow up) of clients [a],[b]** | Number of responses | 74/80 | 85/91 |
| | Weight/BMI | 56 (75.7) | 63 (74.1) |
| | Waist circumference | 25 (33.8) | 30 (35.3) |
| | Waist to hip ratio | 5 (6.8) | 5 (5.9) |
| | Metabolic indicators | 51 (68.9) | 61 (71.8) |
| | Diet improvements | 70 (94.6) | 80 (94.1) |
| | Exercise levels | 52 (70.3) | 60 (70.6) |
| | Quality of life | 51 (68.9) | 60 (70.6) |
| | CBT related changes | 11 (14.9) | 12 (14.1) |
| | Achievement of goals | 55 (74.3) | 62 (72.9) |
| | Clothing size | 25 (33.8) | 27 (31.8) |
| | Blood pressure | 16 (21.6) | 18 (21.2) |
| | Fitness level, aerobic capacity, muscle mass, VO2max | 6 (8.1) | 8 (9.4) |
| | Patient attendance | 20 (2.0) | 23 (27.1) |
| | Patient satisfaction | 34 (46.0) | 40 (47.1) |
| | Medication | 14 (18.9) | 20 (23.5) |
| | Psychological/ body image changes | 24 (32.4) | 28 (32.9) |
| | Other | 5 (6.7) | 5 (5.9) |
| **After an initial consultation/session, how many times the client is reviewed before discharge** | Number of responses | 74/80 | 84/91 |
| | They are not reviewed | 1 (1.4) | 1 (1.2) |
| | 1–2 times | 11 (14.9) | 12 (14.3) |
| | 3–5 times | 16 (21.6) | 18 (21.4) |
| | 6–9 times | 3 (4.1) | 3 (3.6) |
| | 10 times | 2 (2.7) | 3 (3.6) |
| | Other, please specify | 7 (9.5) | 8 (9.5) |
| | Variable/dependent on client needs | 34 (46.0) | 39 (46.4) |

**Table 2.** *Cont.*

| | Completers Only (n = 80) | All Respondents (n = 91) |
|---|---|---|
| | N (%) | N (%) |
| **After an initial consultation/session, the period of time clients would be followed up/reviewed** | | |
| Number of responses | 74/80 | 84/91 |
| Initial consultations given only | 1 (1.4) | 1 (1.2) |
| <2 weeks | 5 (6.8) | 6 (7.1) |
| 2 weeks–1 month | 41 (55.4) | 46 (54.8) |
| 2–3 months | 7 (9.5) | 10 (11.9) |
| 4–6 months | 3 (4.1) | 3 (3.6) |
| 7–12 months | 4 (5.4) | 4 (4.8) |
| >1 year | 2 (2.7) | 3 (3.6) |
| Other | 11 (14.9) | 11 (13.1) |

BMI—body mass index; CBT—cognitive behavioural therapy. [a]—multiple responses allowed and therefore result may be greater than 100%. [b]—n = 85 respondents under all respondents column for this question.

Respondents reported that having access to the following features of their work environment supported their ability to provide effective dietetic treatment to clients with obesity: access to resources (n = 54, 73%), a multidisciplinary team (n = 53, 72%) and length of time available for sessions (n = 53, 72%). The main barriers were time relative to workload (n = 37, 51%) and characteristics of the population group (n = 27, 37%), (Table 3).

**Table 3.** Barriers and Enablers to Practice.

| | | Completers Only | All Respondents |
|---|---|---|---|
| | | N (%) | N (%) |
| **Enablers to practice** [a] | Resources | 54 (73.0) | 60 (74.1) |
| | Access to a multidisciplinary team | 53 (71.6) | 58 (71.6) |
| | Supportive referrers/medical staff | 38 (51.4) | 42 (51.9) |
| | CPD activities done/personal level of experience | 47 (63.5) | 51 (63.0) |
| | Physical environment | 16 (21.6) | 17 (21.0) |
| | Length of time available for sessions | 53 (71.6) | 58 (71.6) |
| | Referral system/mechanism/ access | 29 (39.2) | 32 (39.5) |
| | Patient characteristics or support | 29 (39.2) | 33 (40.7) |
| | Access to data for patient management | 43 (58.1) | 48 (59.3) |
| | Other | 1 (1.4) | 1 (1.2) |
| **Barriers to practice** [a] | Time (relative to workload) | 37 (50.7) | 38 (47.5) |
| | Resources (e.g., referral system, facilities, tools) | 17 (23.3) | 19 (23.8) |
| | Lack of staff (either dietitian or multidisciplinary) | 19 (26.0) | 20 (25.0) |
| | Management related (e.g., not a priority of the service) | 17 (23.3) | 17 (21.3) |
| | Characteristics of the population group | 27 (37.0) | 30 (37.5) |
| | Lack of knowledge with regard to best practice management | 8 (11.0) | 10 (12.5) |
| | Referrer related issues | 14 (19.2) | 16 (20.0) |
| | Lack of skills either psychological, counselling or physical activity | 14 (19.2) | 18 (22.5) |
| | No evidence of treatment effectiveness | 10 (13.7) | 10 (12.5) |
| | There are no barries | 13 (17.8) | 13 (16.3) |
| | Other | 8 (11.0) | 8 (10.0) |

|  |  | Completers Only | All Respondents |
|---|---|---|---|
|  |  | N (%) | N (%) |
| **Gaps in skills limiting the provision of effective dietetic treatment [a]** | Behavioural therapy/modification or psychological assessment or motivation or stages of change | 41 (56.9) | 46 (59.0) |
|  | Knowledge of best practice-guidelines, follow-up, dealing with specific populations | 21 (29.2) | 22 (28.2) |
|  | Resource related, e.g., time, computer, facilities | 15 (20.8) | 17 (21.8) |
|  | Physical activity related | 13 (18.1) | 16 (20.5) |
|  | Client assessment prior to and during therapy, including anthropometric assessment OR goal setting | 8 (11.1) | 8 (10.3) |
|  | Personal interest (lack of) | 10 (13.9) | 10 (12.8) |
|  | No perceived gaps | 8 (11.1) | 8 (10.3) |
|  | Other | 3 (4.2) | 4 (5.1) |

[a]—multiple responses allowed and therefore result may be greater than 100%.

The top two ranked responses related to evaluation of success of treatment were 'through the adoption of improved food and exercise irrespective of weight loss' and 'by the improvement of clinical indicators of health and disease'. Respondents most frequently agreed or strongly agreed that they felt well prepared to treat/manage clients with overweight and obesity (n = 63, 88%). Respondents reported they usually achieved successful outcomes with adult clients (n = 38, 57%). However, responses were more varied when asked if they were professionally prepared to treat or manage children/adolescents with the majority reporting a "neutral" response (n = 15, 43%).

Respondents most commonly reported they always assessed weight history (n = 45, 63%) and exercise habits (n = 47, 66%). The majority of respondents reported they "always" see clients on a one-to-one basis (n = 54, 76%) but mostly "never" utilise a mix of one-to-one and group (n = 51, 72%) consults. When an initial weight management strategy does not work for clients, respondents most frequently reported they "often" or "always" offer another weight management strategy (n = 50, 70%).

The strategies frequently used by respondents are reported in Figure 2 and highly varied. Other frequently used weight loss strategies included other dietary manipulation, e.g., energy density, meal spacing, low glycaemic index and very low calorie diets (n = 63, 90%); cognitive behavioural therapy or review of self-view of body image, self-talk, personal goals and eating enjoyment (n = 53, 76%); and referral to a psychologist for behavioural therapy or stress management (n = 51, 73%).

When asked how they rated their knowledge and level of skill in best practice management of obesity, most respondents rated their knowledge and skill as "good" (n = 35, 47% and n = 33, 45%, respectively). The most frequently identified skill gaps, potentially limiting their ability to provide effective dietetic management were behavioural therapy, modification or psychological assessment or motivation or stage of change (n = 41, 57%), Table 3. The median (IQR) best practice score was 38 (34–44, possible score range 0–62), based on 63 respondent observations. No significant differences were identified between any demographic groups for best practice scores, $p > 0.05$. Cronbach's alpha for best practice score questions was 0.81, which indicated acceptable internal consistency and reliability.

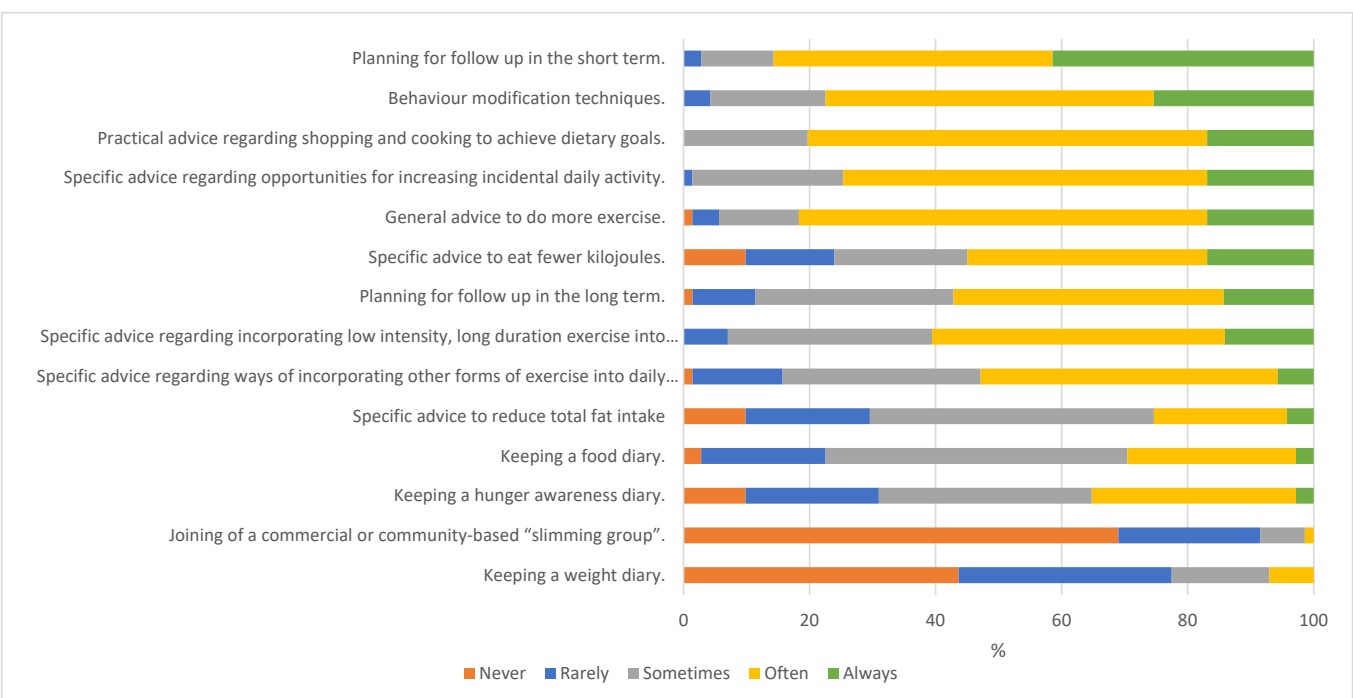

**Figure 2.** Alternative weight management approaches used by survey respondents.

Significant relationships were identified between best practice scores and barriers to obesity management (Supplementary Table S5), these included lower best practice scores by an average of four points for respondents who identified the characteristics of the population as a barrier (F 4.94 (df 1,61), $p = 0.03$), and 6.5 points higher for respondents who reported no barriers (F 7.28 (1,61), $p = 0.01$). Best practice scores were an average four points higher for respondents who reported having access to a multidisciplinary team (F 4.21 (df 1,61), $p = 0.04$) and 4.8 points higher for those who reported CPD activities completed/level of experience as an enabler to practice (F 6.00, (df 1, 61), $p = 0.02$). Differences in best practice scores were significantly lower by an average of seven points for respondents who reported a gap in skills related to client assessment (F 7.19 (df 1,61), $p = 0.01$), and 6.5 points higher scores for respondents who reported no perceived gaps in their skills (F 4.96, (df 1,61), $p = 0.03$). There were no significant differences in best practice scores by year of survey completion.

The majority of respondents had undertaken continued professional development activities to enhance their skills in obesity management, with a preference for counselling skills (n = 56, 76%) and evidence-based practice (n = 51, 69%), with delivery via a webinar/web presentation continued professional development event or a short course most preferred. The most common research questions identified were the effectiveness of dietitian provided interventions (n = 53, 79%), process issues, e.g., frequency of client visits, recidivism, group vs. individual (n = 39, 58%) and non-dietetic issues, e.g., hereditary and surgery (n = 30, 45%).

## 4. Discussion

This survey assessed the use of best practice guidelines, contemporary practice and knowledge and professional development needs of a small sample of Australian APDs working with people with overweight and obesity. It was the fourth implementation of a cross-sectional survey of Australian APDs conducted since 1997.

Dietitians Australia members were invited to participate in the current survey, of the current Dietitian Australia members only 2.3% of financial members took part in the survey with ~1% completing the survey in full [11]. While not all APDs work in this area and there is no data of how many members received the survey invitation, this uptake was

lower than the first three surveys [4–6]. The demographic characteristics of respondents within this current survey were mostly representative of 2020 Dietitians Australia financial members [12], except for having an overrepresentation of respondents who worked >20 h per week and those working in private practice [13].

Only 20% of respondents had accessed or read the Dietitians Australia best practice guidelines, this is fewer than the already low numbers (50%) identified in the 2011 survey [5]. Similarly, only 10% had completed continued professional development activities based on the Dietitians Australia best practice guidelines. Considering almost half of respondents spend 50% or more of their time working with individuals with overweight or obesity, this is a low number of respondents who have accessed the guidelines and continued professional development activities. Additionally, approximately a quarter of respondents did not answer whether they had or had not read or accessed the Dietitians Australia best practice guidelines. The majority of respondents who dropped out of the survey did so in this section, potentially once they may have realised that the survey was asking about their use of guidelines that they may not have accessed. Hence, it is possible that our results were biased towards clinicians who had accessed the guidelines. However, the current best practice guidelines are approximately 10 years old and outdated [8]. It is possible that some respondents may have accessed them previously or that more respondents may seek more up to date evidence sources to support their practice (for example, the National Health and Medical Research Council guidelines), however this information was not captured in the current survey.

General healthy eating advice continues to be the primary dietary approach used by dietitians [4,5], consistent with our current findings. Since the 2002 survey there has been a greater percentage of dietitians using a non-diet approach with behavioural based goals or a non-diet approach that included strategies to achieve reduce energy intake but without counting kilojoules [4]. Use of VLEDs has also increased from 8% (2002) and 18% (2011) to 42% of respondents reporting use of this approach [4,5], this is most likely in line with the growing evidence of the safety and efficacy of VLEDs in weight management [14,15]. The types of strategies used by respondents had not changed since the last survey [5]. However, findings from the current survey showed an increase in respondents taking a patient centred approach and choosing a dietary approach based on patient preference, this increased from 25% and 48% in previous surveys [4,5] to 75% in the current survey. Research shows there is a multitude of strategies which can be adapted for an individual but what is most important is identifying which dietary modifications an individual feels they are most able to adhere to in the long term [16]. Further upskilling APDs in patient centred approaches, as well as medical nutrition therapy, may help APDs to work with patients to identify what approach is going to result in long term success for the patient. The Obesity Canada [9] and European Obesity Guidelines [17] are examples of updated best practice guidelines, with a focus on the patient journey and patient centred care [18]. These guidelines could be used to guide current practice or to inform development of Australian specific medical nutrition therapy guidelines.

Most respondents reported working as part of a multidisciplinary team including general practitioners, psychologists, physiotherapists and exercise physiologists. Further, respondents identified that access to a multidisciplinary team was an enabler to being able to provide effective dietetic treatment to individuals with obesity. Significantly higher best practice scores were identified in respondents who reported having access to a multidisciplinary team as an enabler to their practice. A multidisciplinary team approach has been identified as a necessity to understanding and managing obesity and related diseases [19,20]. Access to a multidisciplinary team helps to provide the long-term support required to promote lifestyle changes in diet, physical activity and health behaviours for the management of obesity [21]. Collaborative approaches to the management of individuals with obesity should continue to be encouraged to further support long-term maintenance. Updated clinical guidelines, should provide recommendations for shared case management, including referral to an APD in order to support optimal health outcomes for patients.

The main barriers to provision of obesity management reported—"time (relative to workload)" and "characteristics of the population"—have not changed since the previous survey in 2011 [5]. Dietitians working in private practice previously reported a lack of time in respect to shorter consult times under chronic disease management plans and time in unpaid administrative work as a challenge [13]. Similarly, other health care professionals have also reported a lack of time as a barrier to holding weight management discussions with patients [22,23]. Having enough time with patients in sessions and follow up was identified as an enabler to providing effective practice and interventions. Additionally, a significantly lower best practice score was identified in dietitians who reported that the characteristics of the population group were a barrier to their practice. Provision of continued professional development activities or practice guidelines that provide strategies to help practitioners hold opportunistic conversations with patients or tips on how to work with individuals with overweight and obesity is warranted to help improve practice. Overcoming these barriers is important as dietitians who reported there are no barriers to their practice in this area had significantly higher best practice scores.

A lack of resources and/or knowledge into how to evaluate practice was identified. At the time of the survey only a third of respondents were currently evaluating the effectiveness of different dietary interventions within their service and only seven respondents had undertaken a recent audit of their obesity service. Forty percent of respondents also expressed interest in continued professional development activities that explained how to evaluate practice and improve evaluation skills. These findings show that dietitians are interested in evaluating practice but may require further training or guidelines. Future guidelines could provide recommendations on how to evaluate practice to help APDs evaluate the effectiveness of interventions in their practice.

Despite the majority of respondents reporting their knowledge of best practice management of overweight and obesity as good or above, best practice management scores were a median of 5–8 points lower in the current survey compared with the previous surveys [4–6]. No significant differences in best practice scores were identified between groups demographic characteristics including between those who had been practicing for longer and those who had more recently started practicing. However, significant differences in score were identified between the type of reported skill gaps. Dietitians who reported a gap in their skills regarding client assessment prior to and during therapy had significantly lower best practice scores. Additionally, dietitians who reported they had no perceived gaps in their skills had an average 5-point higher best practice score. This highlights the need for access to ongoing continued professional development which covers the entire dietetic process starting from assessment. Completing continued professional development in the areas that cover an individuals perceived gaps to practice is important as it is reflective in this cohort of dietitians that those who had greater confidence in their skills also had higher scores aligning with best practice recommendations.

There is increasing evidence available to support a variety of dietary approaches to obesity management [14,16]. However, with conflicting messages between advocating for weight loss versus non-diet approaches there may be growing confusion among APDs on providing interventions. Therefore, updated and promotion of best practice guidelines and continued professional development activities are required to support dietitians to provide effective and safe interventions with patients in larger bodies. Dietitians Australia members have shown that they are interested in further professional development on topics such as counselling skills and evidence-based practice. Ideally the preferred methods of delivery for these continued professional development activities are a webinar/web presentation continued professional development event or a short course.

There are some limitations to the current study. Firstly, the small sample size limits generalisability of findings. Second, there may have been a greater uptake of the survey by respondents who work more than 20 h a week or in private practice due to an increased likelihood of working with individuals with obesity and therefore more interest in this survey. It was not within the capacity of the current survey to recruit a larger sample size,

however future surveys should aim for a larger and more representative sample. Lastly, while questions were asked on use of Dietitians Australia best practice guidelines no further questions were asked about the use of other guidelines respondents may have used to inform practice. Future research could investigate the use of other guidelines by APDs further to identify the primary sources informing evidenced based practice.

## 5. Conclusions

Findings from the current survey, while only in a relatively small sample, suggest that greater uptake and use of best practice guidelines is needed. The current guidelines are almost a decade old and research evidence to support a variety of methods in practice has evolved, such as an increased focus on client centred approaches. Additionally, the low uptake of best practice guidelines needs to be addressed if updated guidelines are to be utilised. Results indicate that additional continued professional development opportunities in areas of expressed need, such as counselling skills, may support and enhance the practice of APDs working with individuals with overweight and obesity. Overcoming the barriers identified to working with individuals with overweight and obesity is important as dietitians who indicated they experienced no barriers when working in this area had significantly higher best practice scores.

**Supplementary Materials:** The following supporting information can be downloaded at: https://www.mdpi.com/article/10.3390/dietetics2010006/s1, Supplementary Table S1: STROBE checklist for the reporting of cross-sectional studies. Supplementary File S2: Survey Questionnaire. Supplementary Table S3: DA Best Practice Guideline Use. Supplementary Table S4: Dietary strategies and monitoring of outcomes used by respondents by year of completion. Supplementary Table S5: ANOVA's comparing Best Practice vs. Enablers, Barriers and Gaps in Practice.

**Author Contributions:** Conceptualization, R.L.H., J.N.B., T.B., L.M.A. and C.E.C.; methodology, all authors; formal analysis, E.D.C.; writing—original draft preparation, E.D.C.; writing—review and editing, R.L.H., J.N.B., T.B., L.M.A. and C.E.C. All authors have read and agreed to the published version of the manuscript.

**Funding:** Clare E. Collins and Tracy Burrows are supported by National Health and Medical Research Council (NHMRC) Investigator Grants.

**Institutional Review Board Statement:** The study was conducted in accordance with the Declaration of Helsinki, and approved by the University of Newcastle Human Research Ethics Committee (H-2020-0212).

**Informed Consent Statement:** Informed consent was obtained from all subjects involved in the study.

**Data Availability Statement:** Data available on request due to restrictions.

**Conflicts of Interest:** The authors declare no conflict of interest.

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
