# Peer review of "Survey of Australian Dietitians Contemporary Practice and Dietetic Interventions in Overweight and Obesity: An Update of Current Practice"

_2674-0311, doi:10.3390/dietetics2010006_

Round 1

Reviewer 1 Report

The article takes up very interesting and more practical topic, it is well written, only a few corrections and requests for additions to the text are included in the authors window.

Abstract

the methodology described in the abstract is not clear - what was the method of collecting data? Please fill in the missing facts

Material and methods

Line 63 The authors wrote “The current survey used similar questions to the previous surveys…” Please clarify and supplement the text what were the differences? if a new tool was created in the form of a questionnaire, it was validated - if so, by what method?

Results

General it is well written., it requires only cosmetic corrections.

·         Tables 1-3 do the selected questions contain asterisks? shouldn't that be removed?

·         Table 1 - not all abbreviations used in the table are explained here, among others for the "responses" category, not all percentages add up to 100

·         Some questions are a consequence of others (related to a positive answer) - please clarify whether they were asked only when the respondents answered YES to the preceding question - is it not clear?

For example: “Do you work with other members of a multidisciplinary team?” Yes No and next question is „What other services are provided within your multidisciplinary team?

And the next example „Are you a member of any obesity interest groups (DA or other)? No (n=47) Yes (n=28) and the next question „What obesity interest groups are you apart of? all people who answered Yes were 28) (previously there were 27 positive answers?) how is this possible?

And the next example „Does your service use the clinical guidelines for obesity management? Yes No Unsure”. The positive answers were 40 why in the next related question („What year did your service start using the clinical guidelines for obesity management?”) only 20 people replied?

in the methodology and in the tables, it is necessary to clearly describe which questions were single choice and which were multiple choice.

Discussion

Line 272 “characteristics of the population - have not changed since the previous survey in 2011” the authors indicate that the studied population did not differ from those in previous years - how was this lack of difference assessed? on what basis was this found? After all, 11, 20 and 25 years have passed compared to the previous studies

Author Response

Thank you for your comments, please see our responses in the attached document. 

Reviewer 2 Report

Thank you for the opportunity to review the manuscript titled “Survey of dietetic intervention in overweight and obesity: an 2 update of current practice” by Clarke et al.

This survey assessed the use of best practice guidelines, contemporary practice and knowledge and professional development needs of a small sample of Australian APDs working with people with overweight and obesity. It was the fourth in a series of surveys 215 of Australian APDs conducted since 1997. All DA members were invited to participate in the current survey, of those invited only 3% of financial members took part in the survey with ~1.3% completing the survey in full. This study is a descriptive cross-sectional study.

Major

1.     The title needs improvement. Current title is “Survey of dietetic intervention in overweight and obesity: an update of current practice.”. Need to be more clear.

2.     Small sample size – Authors have acknowledged this as a limitation. However, is it adequate? This is much smaller than surveys done in previous years. It is not possible to generalize with this small sample size.

3.     The exact response rate was not clear since in the discussion it mentions that the survey was sent to all dietitians. However, in another place it mentions that “the survey was attempted by 178 participants, but the response rate was 45% (n=80).” Be clear about the # at each stage.

4.     In the analysis, it was not controlled for the year of data collection – Since this was during the COVID-19 pandemic, there could have been an impact on the data. As time progressed with the pandemic more and more information was available and the association between severe disease and obesity became more and more known. This factor could have impacted the dietitians attitudes and behavior by 2021. – Either a supplementary analysis controlling for the data collections years should be done or the authors should highlight this as a limitation.

5.     Analysis of group differences – suggest checking for normality and then deciding on whether to use parametric or non parametric .

6.     Table 1 & 2 - The authors considered looking into see if there were any differences between the responders and non-responders. However, the Table 1 and 2 shows completers and all responders. Suggest having completers (n=80) and non-completers (n=98) for a better understanding and comparison. Be clear what is meant by non-completers. Is it those who did not answer the question on alignment with the guideline? Add another column to indicate the chi-square p- values. Since some of the groups had 0 values, Fisher’s test may be needed for those.

7.     Conclusion –The current content in the conclusion is more of a future direction description and some of it needs to be moved up to the discussion section. Thus, the conclusion needs to be re-written. Authors need to summarize their main findings and provide a conclusion based on that.

8.     Authors did not assess internal consistency of their survey questions, that were used to calculate best practices score, using Cronbach’s alpha. Assessing the validity of a survey is important to ensure that they obtained meaningful data. 

9.     This paper is only reporting frequencies and authors could have studied associations using the best practices scores.

10.  Highly recommend the authors to use the STROBE guideline for reporting observational studies to improve the quality of the paper. Following that check list will improve all aspects of reporting and organization of this manuscript. https://www.equator-network.org/reporting-guidelines/strobe/. Using this will help to address several of the above-mentioned points. This could be attached as a supplement.

Minor

·       Table 1 & 2– Indicate the number of responders for each question. For example, it looks like only 75/ 80 answered the question #1 in Table 1

·       Table 1- All abbreviations should be explained including those for the question “ Which DA Branch do you be-long to?”

·       Suggestion using fewer abbreviations because it can get confusing for the reader. While the abbreviations “APD, PA, CPD may commonly be used in Australia, other international readers may have a hard time remembering these as they read the article.

·       Writing needs to be improved and some sentences needs to be reworded.

·       Suggest adding a table to summarize the scores and including the # of observations (n) considered per category. In that table have a column for non-respondents and indicate the group difference p value too.

·       Figure 1 – provide axis details

·       Is Ref#10 appropriate? It is not a peer reviewed article, but is a media release. The link does not work .

Line 23            Statement is not clear- It could be reworded as “seventy five percent of the dietitians reported that  dietary intervention were selected ………….”

Similarly two next line could be reworded to start off with the percentage of dietitians.

Line 26            What is meant by “uptake”? Better to summarize the findings and reword in conclusion in the abstract. At present only part of the findings are summarized as conclusion.

Line 28            It would be better to say “may be helpful” instead of “required to”.

Line 45 – 46    This sentence is a bit confusing, especially “update survey findings”.  “Given recent research development and potential impact on practice, there is a need to update survey findings.” Reword.

Line 57            A comma is needed after the word “towards”,

Line 59            add “the” in front of “use of the best ……”

Line 63            There is an extra space between the words “following” and “three”.

Line 66            Mention what year the DA guideline is.

Line 73            Describe what “DA and Dietitian Connection” are. What kind of organization?

Line 100          “Results were provided for completers only and for all respondents.” – better to delete “for all respondents”. Is there a rationale for keeping it?

Line 127- 128 “Approximately 127 half of respondents (56%) were not involved in any obesity interest groups; among those 128 most commonly reported was the DA obesity interest group.”- this statement is not clear. Suggest making it two sentences.

Line 131          Avoid using abbreviations in the titles or subtitles

When discussing the results, refer the table # so that the reader will know where to look for the results.

Line 201, the findings for all participants were mentioned without mentioning what group it is.

Thus, when reporting/ discussing data in the text in the Results section, recommend mentioning what group is being mentioned.

Line 204          Indicate the p value.

Line 248          indicate how much the increase was from 2011

Line 84            Add where these questions are found in the supplements.

Line100           Add subtitle “Statistical analysis”

Line 203          Add “for” before “best practices scores”.

Author Response

(The authors gave the same response as above.)

Round 2

Reviewer 1 Report

 thank the authors for introducing corrections according to my suggestions, in this version the work is much better, and in my opinion it is ready for publication

Author Response

Thank you for reviewing our manuscript. 

Reviewer 2 Report

Thank you for addressing the comments. Few minor things to edit. 

Line 17 – Add a comma after the word “identify”.

Where is supplementary table 2? There is 1 and 3 but 2 is missing in the supplementary material doc.

Is “S1 Survey” document the Supplementary 2? Needs to be labeled correctly.

Line 173 – Is “respondents” or completers? 

Author Response

Thank you for your comments, please see attached our responses. 
